# Baseline Electrocardiographic and Echocardiographic Assessment May Help Predict Survival in Lung Cancer Patients—A Prospective Cardio-Oncology Study

**DOI:** 10.3390/cancers14082010

**Published:** 2022-04-15

**Authors:** Sabina Mędrek, Sebastian Szmit

**Affiliations:** 1Department of Cardiology, Subcarpathian Oncological Center, 36-200 Brzozów, Poland; sabp@wp.pl; 2Department of Pulmonary Circulation, Thromboembolic Diseases and Cardiology, Centre of Postgraduate Medical Education, European Health Centre, 05-400 Otwock, Poland

**Keywords:** cardio-oncology, lung cancer, performance status, cardiovascular comorbidities, ECG, atrial fibrillation, echocardiography, strain rate, survival, prognosis

## Abstract

**Simple Summary:**

Lung cancer is characterized by a high mortality rate. The main prognostic factors are histopathological and molecular diagnosis as well as clinical advancement. Performance status and comorbidities have an additional impact on the outcome. Assuming that lung cancer is often associated with cardiovascular diseases and the localization of the disease in the chest leads to exercise dyspnea, it seems valuable to objectively determine what echocardiographic parameters correlate with significantly reduced (ECOG = 2) performance status. Some echocardiographic parameters (RVSP, ACT, RV GLS, RV free wall strain, fractional area change) were associated with low performance status which can help to objectively confirm exercise intolerance in lung cancer. Moreover, recognition of atrial fibrillation and high heart rate (>90/min) in ECG strengthens the prediction of earlier death. Meanwhile, finding some positive prognostic factors like RVSP < 21 mmHg or RV free wall strain < −30% and even RV GLS < −25.5% in echocardiography allows planning of more aggressive anticancer treatment.

**Abstract:**

Cardiovascular disease and cancer coexist and lead to exertional dyspnea. The aim of the study was to determine the prognostic significance of cardiac comorbidities, ECG and baseline echocardiography in lung cancer patients with varying degrees of reduced performance status. This prospective study included 104 patients with histopathologically confirmed lung cancer, pre-qualified for systemic treatment due to metastatic or locally advanced malignancy but not eligible for thoracic surgery. The patients underwent a comprehensive cardio-oncological evaluation. Overall survival negative predictors included low ECOG 2 (Eastern Cooperative Oncology Group) performance status, stage IV (bone or liver/adrenal metastases in particular), pleural effusion, the use of analgesics and among cardiac factors, two ECG parameters: atrial fibrillation (HR = 2.39) and heart rate >90/min (HR = 1.67). Among echocardiographic parameters, RVSP > 39 mmHg was a negative predictor (HR = 2.01), while RVSP < 21 mmHg and RV free wall strain < −30% were positive predictors (HR = 0.36 and HR = 0.56, respectively), whereas RV GLS < −25.5% had a borderline significance (HR = 0.59; *p* = 0.05). Logistical regression analysis showed ECOG = 2 significantly correlated with the following echocardiographic parameters: increasing RVSP, RV GLS, RV free wall strain and decreasing ACT, FAC (*p* < 0.05). Selected echocardiographic parameters may be helpful in predicting poor performance in lung cancer patients and, supplemented with ECG evaluation, broaden the possibilities of prognostic evaluation.

## 1. Introduction

Current research has identified many common pathomechanisms underlying the development of heart failure (HF) and cancer [1]. Inflammation is one of the common denominators [2]. Furthermore, cancer is increasingly common in patients already diagnosed with HF [3]. HF seems to predispose to the development of cancer [4].

The coexistence of cancer and cardiovascular disease is becoming more frequent [5]. Common risk factors for developing cancer and cardiovascular diseases include—at least—older age, metabolic disorders, an unhealthy lifestyle including, among others, smoking or exposure to other toxins, a sedentary lifestyle with exercise avoidance [6,7,8].

Lung cancer, regardless of whether non-small cell lung cancer (NSCLC) or small cell lung cancer (SCLC), is an example of a cancer which is very often accompanied by cardiovascular diseases [9]. Up to 50% of lung cancer patients may have concomitant cardiovascular disease at baseline [10]. Available data show that accompanying cardiovascular diseases shorten survival time in both SCLC and NSCLC [11,12].

On the other hand, cancer may be considered as a potential risk factor for HF [13]. Anticancer treatment poses an additional risk of inducing new cardiovascular diseases, including HF [14]. This applies not only to systemic treatment but also to radiotherapy [15].

Each advanced cancer, especially lung cancer, is beginning to be understood by some investigators as one of the manifestations of symptomatic HF [16]. The observed ventricular arrhythmias are one of the clinical rationales [17]. Their prognostic significance has been demonstrated, among others, in lung cancer [18]. Physical activity appears to play an extremely important role in cancer as it not only reflects daily quality of life, but its improvement may determine a more favourable survival time [19]. Lung cancer with all the limitations of ventilation reduces cardiopulmonary efficiency. In oncology, performance status, usually assessed according to ECOG (The Eastern Cooperative Oncology Group scale), is also based on the assessment of exercise tolerance. ECOG 2 seems problematic in terms of qualification for anticancer treatment because it reflects the ability to perform personal activities, but inability to work. Patients with ECOG 2 spend about half of the day in bed.

Each lung disease, including lung cancer, similar to heart failure is characterized by different levels of dyspnea. There are different definitions of heart failure. The definition proposed by the European Society of Cardiology described heart failure as a complicated clinical syndrome consisting of symptoms of dyspnoea or fatigue occurring at rest or during exercise, which are caused by structural or functional cardiac abnormalities [20].

Lung cancer, due to the nature of its specific risk factors, location and clinical course, as well as concomitant cardiovascular diseases, appears to be a challenge in modern cardio-oncology. The location of the tumour itself can cause cardiorespiratory failure. Additionally, systemic treatment can induce many different cardiovascular complications [21]. Cardio-oncology experts suggest a thorough cardiological evaluation before the onset of cardiotoxic anticancer therapy [22]. However, the document does not mention physical activity, exercise tolerance or the role of baseline echocardiography in predicting prognosis related to the cancer itself, which seems of particular importance in lung cancer. An open question at this point is how to interpret echocardiographic abnormalities found in lung cancer. 

The aim of this study was to determine the prognostic significance of cardiovascular comorbidities, ECG and baseline echocardiography in lung cancer patients with varying degrees of exercise intolerance/reduced performance status.

## 2. Materials and Methods

### 2.1. The Main Design of the Study

This prospective study was planned to include patients with histopathologically confirmed lung cancer, pre-qualified for systemic treatment for either (1) metastatic cancer or (2) locally advanced cancer not eligible for surgery.

The study was designed as a pilot study to test the potential relationship between overall mortality and echocardiographic or electrocardographic changes related to advanced lung cancer disease. The idea of the study was to check whether echocardiographic and electrocardiographic parameters enhanced by lung cancer development could complement the prognostic risk scales in inoperable advanced or metastatic lung cancer disease.

The study was conducted in one cancer center that specializes in treating only advanced/metastatic lung cancer disease. Due to the fact that the center does not have thoracic surgery, patients with indications for surgical treatment were referred to another center. Each patient enrolled in the study had to be disqualified from lung cancer surgery by a multidisciplinary team.

The objective endpoint of the study was all-cause mortality; therefore it was possible to include patients with various histopathological and molecular diagnoses.

It was planned to include at least 100 patients so that the demonstrated relationships become the basis for creating a study dedicated to particular types of advancement (III vs. IV) and typical histopathological (at least SCLC vs. NSCLC) and a specific treatment (radical chemoradiotherapy in locally advanced disease, immunotherapy and chemotherapy in metastatic disease, etc.).

### 2.2. Diagnostics Tools in the Study

The patients included in the study underwent a comprehensive cardio-oncological assessment including detailed history, physical examination, laboratory tests, electrocardiography (ECG) and echocardiography.

Oncological history included the Eastern Cooperative Oncology Group (ECOG) performance scale assessed by the attending oncologist, tumour staging classification (TNM), histopathological examination, amount of pain medication used, and cardiological history including hypertension, chronic coronary syndrome, cardiac arrhythmias, other comorbidities (diabetes, kidney disease, hypothyroidism, chronic obstructive pulmonary disease) and medications used.

During the visit, vital signs, height, weight, body mass index (BMI), body surface area (BSA) and blood pressure were measured.

The laboratory parameters analysed included blood count, creatinine, potassium levels, liver tests, bilirubin, TSH and lipid profile.

Resting electrocardiogram (ECG) was performed using an AspelAsCARD Grey 07.305P device to assess heart rate and rhythm, PQ interval, corrected QT interval (QTc) and QRS complex (i.e., mainly the presence or absence of right bundle branch block-RBBB).

Resting transthoracic echocardiography (TTE) with simultaneous ECG was the key diagnostic element. A Philips HD 15 was used for the examination and all images were digitally recorded on a hard drive in DICOM format for subsequent off-line analysis. All examinations were performed by the same echocardiographer. According to the recommendations of the American Society of Echocardiography (ASE) and the Echocardiography Section of the Polish Cardiac Society [23,24] the following measurements were performed:(1)Parameters assessing morphology, left ventricular systolic and diastolic function:
−left ventricular end-diastolic dimension (LVEDD);−left ventricular end-systolic dimension (LVESD);−ventricular septal diastolic dimension (IVSDd);−posterior wall diastolic dimension (PWDd);−left atrial area measured in the 4-chamber view (LAA);−left ventricular ejection fraction (LVEF) measured by Simpson’s method;−global longitudinal strain (GLS);−ratio of maximum velocities of wave E to wave A (E/A);−early diastolic mitral annular velocity (E’);−ratio of maximum early mitral inflow velocity to end-diastolic mitral annular velocity (E/E’)
(2)Parameters assessing right ventricular structure and function and pulmonary hypertension:
−right ventricular end-diastolic dimension (RVEDd);−right atrial area measured in the 4-chamber view (RAA);−global longitudinal strain (GLS);−systolic longitudinal strain (GLS) of the right ventricular free wall;−tricuspid annular motion amplitude (TAPSE);−diameter of the inferior vena cava (IVC);−pulmonary artery pressure (RVSP).
(3)The following parameters were additionally assessed:
−additional structures in the heart cavities;−transvalvular flow to exclude significant valvular defects;−pericardial fluid.


### 2.3. Endpoint in the Study

The main endpoint of the study was all-cause mortality. 

All the above parameters were related to overall survival (OS), defined as the time from the onset of anticancer treatment to death from any cause. 

The study received approval from the Bioethics Committee No. 236/KBL/OIL/2018 on 11 December 2018. The first patient was included in the study on 23 January 2019. The follow-up lasted until 2 November 2021. 

## 3. Statistical Analysis

All nominal parameters were defined in terms of frequency, parameters on a continuous scale were defined by the arithmetic mean (for normal distribution) or by the median and quartiles (for a distribution other than normal). The Cox proportional-hazards model was used to determine the prognostic significance of all evaluated oncological and cardiovascular parameters. Lower and upper quartile values were used to present the prognostic significance of echocardiographic parameters. Logistic regression analysis was used to determine the association between limited performance status and echocardiographic parameters. All statistical analyses were performed using STATISTICA software.

## 4. Results

The group was heterogeneous in terms of histological tumour type (Table 1). Stage IV malignancy (metastatic) predominated among patients (55.8%, *N* = 58): 42 patients (40.38%) with non-small-cell lung cancer (NSCLC) and 16 patients (15.38%) with small-cell lung cancer (SCLC).

According to the inclusion criteria, all patients had advanced or metastatic lung cancer disease. Each patient had an initial CT scan to objectively assess the advancement of the neoplastic disease. The tomography showed none of the patients had myocardium or pericardium directly affected by the neoplasm, which would have resulted in the prognosis being much worse due to the direct involvement of the heart structures.

Qualification for lung cancer treatment was based on the histopathological and molecular diagnosis, taking into account the clinical stage, the performance status and comorbidities including cardiovascular diseases. Patients received anticancer treatment according to the current guidelines in Poland [25]. Finally, the patients received at first choice:➢platinum-based doublet chemotherapy: 72 patients (69.23%)
○cisplatin-based: 53 patients (50.96%)
▪cisplatin + vinorelbine: 28 patients▪cisplatin + etoposide: 14 patients▪cisplatin + pemetrexed: 6 patients▪cisplatin + gemcitabine: 5 patients
○carboplatin-based: 19 patients (18.27%)
▪carboplatin + etoposide: 10 patients▪carboplatin + vinorelbine: 8 patients▪carboplatin + paclitaxel: 1 patient

➢cytotoxic monotherapy: 17 patients (16.35%)
○pemetrexed: 6 patients○vinorelbine: 5 patients○carboplatin: 3 patients○gemcitabine: 2 patients○taxanes: 1 patient
➢monotherapy with pembrolizumab: 8 patients (7.69%)➢targeted therapy: 3 patients (2.88%): two with EGFR inhibitor and one with ALK inhibitor➢alone radiotherapy: 2 patents (1.92%)➢best supportive care: 2 patients (1.92%).

Additionally, six patients (5.77%) received radiotherapy due to locally advanced NSCLC as part of chemoradiotherapy (concomitant in two patients, sequential in four patients). The next six patients (5.77%) were treated by radiation therapy due to metastases to bone or central nervous system. 

In the long-term follow-up when lung cancer disease progression was recognized a second line of cancer therapy was recommended:➢chemotherapy: twelve patients (11.54%) including: platinum-based scheme in six patients, gemcitabine in three patients, taxanes in two patients and CAV chemotherapy (cyclophosphamide, doxorubicin and vincristine) in one patient➢immune checkpoint inhibitors: four patients (3.85%) including: atezolizumab in two patients, nivolumab in one patient, pembrolizumab in one patient➢radiation therapy in 33 patients (31.73%): for lung in 16 patients, for central nervous system in nine patients, for mediastinum in five patients, for other site of metastasis in three patients.

During the study observation period, 87 patients (83.65%) died. There were no clearly confirmed cardiovascular reasons for death. At the end of follow-up 17 patients (16.35%) were alive. Median of overall survival (OS) was 260 days (8.67 months) with interquartile range: 71–623 days (2.37–20.77 months). The 1-year and 2-year survival rates were 39.42% and 17.84%, respectively (Figure 1).

The significant advancement of the lung cancer disease (inoperable stage) resulted in a high mortality (Figure 1). During the first 100 days of observation, 19 patients (27.88%) died due to cancer-related reasons. The remaining 75 patients underwent electrocardiographic and echocardiographic control. Cardiovascular toxicities of lung cancer therapies were identified according to the International Cardio-Oncology Society 2021 consensus statement (Table 2) [26]. Cardiac toxicity was diagnosed in 24 of 75 patients (32%): as severe in four patients (5.33%), moderate in the next four patients (5.33%), mild in 16 patients (21.33%).

There were only a few clinical cardiovascular events related to anticancer therapy. Atrial fibrillation can be observed in response to anticancer therapies, but only one patient, a 75 year old woman with history of arterial hypertension and chronic pulmonary disease, with low performance status ECOG 2, experienced atrial fibrillation during treatment with vinorelbine due to lung adenocarcinoma (Table 2). All cases of arterial hypertension were recognized prior to the diagnosis of lung cancer. After the initiation of anticancer treatment, there was no new diagnosis of arterial hypertension, but there were seven patients who were discontinued ACEI or ARB due to the observed hypotension. Only two new venous thromboembolic events were recognized: one in a patient treated with pembrolizumab, and another one in a patient receiving chemotherapy with cisplatin and vinorelbine. There were no diagnoses of acute heart failure or acute coronary syndromes.

All baseline oncological and cardiac data were correlated with survival time (Table 3). Histopathological diagnoses were distributed in such a way that they were not significantly associated with survival time. Among the oncological history data, low ECOG 2 performance status, stage IV malignancy i.e., metastases, especially bone or liver/adrenal metastases, pleural effusion, were OS predictors. Analgesics, both the amount and the need for morphine, were also important prognostic factors. Among cardiological factors, two ECG parameters (atrial fibrillation and heart rate above 90/min) were found to be important.

Among the echocardiographic parameters, the following were found to be significant for prognosis (Table 4): high RVSP (>39 mmHg) as a negative predictor and low RVSP (<21 mmHg), RV free wall strain (<−30%) as positive predictors, and RV GLS (<−25.5%) with borderline significance.

Logistical regression analysis showed that ECOG = 2 was significantly correlated with the following echocardiographic parameters (Table 5): RVSP (OR = 1.03 for each 1 mmHg), ACT (OR = 0.97 for each 1 ms), RV GLS (OR = 1.14 for each 1%), RV free wall strain (OR = 1.12 for each 1%), Fractional Area Change -FAC (OR=0.87 for each 1%).

## 5. Discussion

Lung cancer is one of the most commonly diagnosed cancers, especially in men, and at the same time the leading cause of cancer-related deaths worldwide in both men and women. The 5-year survival rates for all stages of lung cancer do not exceed 15–20% [27]. Factors affecting the prognosis include tumour size and differentiation, gender, age, smoking, general fitness status, comorbidities, type of lung resection and experience of the treating centre [28]. Tumour progression based on TNM classification has a significant influence on prognosis. This classification has changed over the years; however, the division into localised (involving the lung), regional (involving lymph nodes) and metastatic cancer remains unchanged. 

We know from a 1997 publication that the 5-year survival of patients with non-small cell lung cancer (NSCLC) was classified according to stage: 61% for IA, 38% for IB, 34% for IIA, 22–34% for IIB, 1–8% for III, and only 1% for stage IV [29]. A 2007 analysis reported data indicating that the 5-year survival among 67,725 NSCLC patients from different European countries was 50% for IA, 40% for IB, 24% for IIA, 25% for IIB, 18% for III A, 8% for III B, and 2% for IV [30]. Based on the SEER registry including patients with newly diagnosed lung cancer between 2010 and 2016, the 5-year survival for NSCLC patients was 63% for localized, 35% for regional, and 7% for metastatic; and among SCLC (small cell lung cancer) patients: 27% for localized, 16% for regional, and 3% for distant; as well as 25% in NSCLC and 7% in SCLC for all SEER stages combined [31]. The CONCORD-3 study covering 37.5 million cancer patients diagnosed between 2000 and 2014 found that the 5-year survival in patients with lung cancer in Poland was only 14.4%, mainly due to the diagnosis at an advanced stage, i.e., III B or IV (60–85%) [32].

Patients within each clinical stage still represent a heterogeneous group, requiring further differentiation. Histological type is an independent predictive factor favourable for NSCLC compared to SCLL. No complete consensus has been reached in clinical trials for the assessment of prognosis within the different histological types of NSCLL. Grossi et al. found histological type to be an independent prognostic factor; the 5-year survival was significantly higher (60%) for squamous cell carcinoma compared to non-squamous cell carcinoma (49%) [33]. Hubbard et al. showed that the diagnosis of squamous cell carcinoma in operated patients was associated with worse overall survival than adenocarcinoma, but with better disease-related survival time [34]. The histological grade, which is classified as highly differentiated, moderately differentiated, poorly differentiated and undifferentiated, is an important aspect. Histological grade was found to be a significant independent prognostic factor [35]. The degree of histological differentiation is related to the amount and duration of exposure to tobacco smoke. High exposure is associated with less differentiated and more aggressive tumours, and thus lower survival regardless of comorbidities and other factors. In a multivariate analysis, Kuo et al. showed no impact of histological type on OS in stage I patients; histological differentiation and CEA levels had the greatest impact on prognosis [28]. In our research, we were able to select the study population such that the histopathological diagnosis was not significantly associated with OS, which facilitated further analysis on the significance of cardio-oncological parameters.

The patient’s general condition and quality of life as assessed by the Eastern Cooperative Oncology Group performance status scale (ECOG scale) is another factor with prognostic value. This five-point scale runs from asymptomatic, able to perform daily activities independently (a score of zero) to death (a score of five). ECOG is one of the strongest prognostic indicators, along with disease stage and weight loss in the 6 months before cancer diagnosis [36]. The following are favourable prognostic factors: good performance status (ECOG 0.1), normal body weight or its slight loss (≤5% of normal value); in case of NSCLC, these are female sex and the absence of *KRAS* mutations [37]. Most clinical trials divide patients into ECOG 0–1 and ECOG 2 groups. When comparing these two groups, shorter median survival, regardless of treatment (PS 0–1: 6.4 months, PS ≥ 2: 3.3 months) and shorter one-year survival (20% and 9%, respectively) is clearly seen in advanced NSLCL [38]. Indeed, it becomes a clinically relevant question whether ECOG ≥ 2 can be treated as the specific equivalent of cardiopulmonary failure or even heart failure itself in lung cancer, with such a specific location of the tumour in the thoracic region and multiple concomitant cardiovascular diseases. Our work shows that ECOG = 2 clearly correlates with some echocardiographic parameters. Thus, significant lung cancer symptoms correlate with structural or functional abnormalities on echocardiography. 

The presence of comorbidities at the time of diagnosis significantly worsens prognosis, which is associated with both a delay in lung cancer diagnosis and limitations in the use of surgical and systemic treatment. The Charlson Comorbidity Index (CCI) is the most commonly used tool to estimate the probability of death within a year for patients with comorbidities. A score of ≥3 is associated with an 80% increase in the risk of death within a year [39]. Chronic obstructive pulmonary disease (COPD) is the most common comorbidity in lung cancer. The prevalence of coexistence of these two diseases is estimated at 52% [40]. COPD is a risk factor for lung cancer and worsens its prognosis. According to Iachin et al., COPD reduces 5-year survival by 20%in patients with NSCLC [41]. Cardiovascular diseases, including coronary artery disease, hypertension, arrhythmias, and peripheral arteriosclerosis, also worsen the prognosis by increasing the risk of death by 30% compared to patients without these conditions. Mortality is also affected by diabetes, which increases mortality by 20%, as does cerebrovascular disease. In our population, no direct effect of any of comorbidities on OS was demonstrated. This was probably due to the fact that the vast majority of patients had advanced cancer, but also to the fact that comorbidities were present in a high percentage of patients.

There is an ongoing search for clinical and molecular prognostic and predictive indicators in lung cancer. There is also a clinical need to develop non-invasive diagnostic tools with additional prognostic significance that can modify treatment eligibility and improve the prognosis of cancer patients. Objective, reproducible prognostic imaging parameters are sought. Transthoracic resting echocardiography, which seems to be underestimated in lung cancer, may be one of the non-invasive diagnostic methods useful in prognosis. 

Echocardiography is usually used in oncology to exclude severe heart disease disqualification from anticancer treatment [42]. Another indication is the diagnosis of cardiotoxicity from anticancer therapy [26]. This diagnosis is largely based on the assessment of left ventricular ejection fraction (LVEF) and left ventricular global longitudinal strain (GLS). There are few studies on right ventricular function for both prognostic and cardiotoxicity assessment. Non-invasive assessment of right ventricular function by echocardiography has been attempted in recent years. Tadic et al. [43], who compared patients with solid tumours before anticancer treatment versus controls, found lower right ventricular GLS in the group of oncological patients, whereas other parameters (right ventricular free wall GLS, RVSP, TAPSE, right ventricular dimension) were similar. Chen et al. [44] demonstrated a decrease in right ventricular GLS and right ventricular free wall GLS 6 months after chemotherapy in patients with stage III NSCLC. Other right ventricle parameters remained unchanged. A clear relationship was demonstrated between decreased right ventricular free wall GLS and mortality. The mechanism underlying right ventricular damage remains unclear; according to the authors, it was mainly related to radiotherapy. A study to assess right and left ventricular function was also performed in breast cancer patients undergoing trastuzumab treatment (after prior anthracycline treatment or during taxane therapy). Keramida et al. [45] demonstrated reduced left and right ventricular GLS with similar values, with a maximum drop at 6 months of treatment; however, data on the effect on mortality is missing. Demonstration that right ventricular GLS and right ventricular free wall GLS may be predictors of good prognosis in lung cancer is the unique clinical value of our work. It is worth noting that the abnormalities of these parameters were determined prior to the initiation of systemic anticancer treatment. Thus, these prognostic echocardiographic parameters may be markers of complications of co-morbidities, although they probably mainly reflect lung cancer advancement and affected sensitive right ventricular parameters.

Right ventricular systolic pressure (RVSP) is another echocardiographic parameter assessing right ventricular function. In historical publications, echocardiographic measurement of the maximum gradient obtained from tricuspid return wave velocity has been shown to correlate with pulmonary artery systolic pressure values measured invasively [46]. More recent studies use echocardiographically measured RVSP in predicting pulmonary artery pressure in various clinical situations, even as difficult as the assessment of sarcoidosis patients at risk of pulmonary hypertension [47]. RVSP has been shown to be useful in the prognostic assessment of patients in cardiac intensive care units [48]. Kjaergaard et al. [49], who assessed non-cancer patients with both preserved and reduced left ventricular ejection fraction, showed that each 5 mmHg increase in right ventricular systolic pressure increased mortality by 9% over a 5.5-year follow-up. In our study in patients with lung cancer, high RVSP clearly correlated with higher mortality, while low values were prognostically favourable. It seems that in our population RVSP is also an exponent of tumour progression and therefore probably influences prognosis. Due to the special nature of our population it seems reasonable to continue these analyses in future studies.

Resting electrocardiogram is another non-invasive diagnostic test. The assessment of the presence of cardiac arrhythmias, especially atrial fibrillation (AF), is an important parameter. AF is the most common arrhythmia, affecting 2–4% of the general population. Its frequency increases with age (over 85 years of age, 36% of patients are affected). AF may be asymptomatic and is detected incidentally in up to 30% of patients during standard or continuous ECG recording [50,51]. The prevalence of AF in patients with cancer is higher than in the general population, approximately 20%, and depends largely on the type of cancer and the treatment used [52]. The most common cancers in the population are associated with AF risk [53]. The incidence of newly diagnosed AF (after diagnosis) is associated with a higher tumour grade and thus a worse prognosis and higher cardiovascular mortality [54,55]. AF increases the risk of cancer, lung and gastrointestinal malignancies in particular [56,57,58,59]. Atrial fibrillation is a typical complication of thoracic surgery [60,61]. To date, a worse prognosis has been demonstrated in patients undergoing thoracic surgery for lung cancer who develop atrial fibrillation (HR for mortality 3.8) [62]. Data on increased overall mortality in patients with lung cancer and pre-existing AF on systemic anticancer therapy are missing. Our study is the first to document the significant negative prognostic significance of AF in patients with lung cancer eligible for systemic therapy.

AF often coexists with heart failure (HF) both with reduced and preserved left ventricular ejection fraction (EF). AF increases the risk of death, HF hospitalisation, stroke and TIA regardless of EF values in all types of HF [63]. The mechanisms leading to an increased incidence of AF in cancer patients are not fully elucidated. In addition to classical risk factors present in the general population (hypertension, diabetes, etc.), tumour-related factors, i.e., water-electrolyte disturbances, hypoxia, sympathetic overactivity due to pain, emotional stress and cancer treatment-related factors, are taken into account.

Sinus tachycardia is defined as a resting sinus rhythm rate above 100/min. Many studies indicate that an elevated resting heart rate is an independent risk factor for death [64,65]. Mohamad et al. showed that sinus tachycardia occurring during cancer treatment is associated with an increase in cardiovascular events and mortality over a 10-year period [66]. In our study, the prognostic significance of heart rate was investigated regardless of whether it was sinus rhythm or atrial fibrillation. It was found that heart rate above 90/min already predicts a higher mortality risk. It can be concluded that this is another functional parameter correlated with cancer advancement and a worse prognosis.

Pathophysiologically, our results can be explained as follows. Neoplastic or inflammatory infiltration in the lung leads to hypoxia, vascular remodelling and increased vascular resistance, which in turn increases right ventricular pressure, causing pressure overload and a secondary decrease in right ventricular GLS and FAC. Hence, these parameters correlate with ECOG 2 clinical symptoms in our population. Furthermore, RVSP increases with lung tumour progression, resulting in poorer contractility of the interventricular septum, and hence both right ventricular and free wall GLS get worse. Thus, clinically advanced lung cancer may resemble terminal heart failure, with classical signs of right ventricular failure. Therefore, a potential explanation is that a less negative value of right ventricular and right ventricular free wall GLS is indicative of earlier death in lung cancer patients.

The small sample size with heterogeneous histopathological diagnoses and clinical stages is an important limitation of the study. However, it should be noted that the distribution of histopathological diagnoses corresponds with the global epidemiology of lung cancer, and this was a “preliminary one centre study”. Thus, despite the small size of the study group, the analysis reflects everyday practice by showing how lung cancer varies in terms of oncological characteristics, but also in terms of concomitant internal medicine problems, including cardiac ones. However, it is important to notice that due to the small size of this pilot study, the echocardiographic parameters have not been applied to age and gender.

Identification of good prognosis parameters, which could facilitate qualification for more intensive cancer treatment in many clinical situations, is an asset of this study. Qualification of elderly patients for concurrent chemoradiotherapy, where sequential treatment or radiotherapy alone is a commonly considered alternative option, is a classic example [67]. The presence of favourable predictive exponents in echocardiography and more precise ECOG determination after taking into account echocardiographic parameters, could help in such qualification; however, further research is needed. Another difficult clinical issue is the qualification of patients with limited cardiopulmonary function in the course of cancer alone and concomitant diseases for thoracic surgery. It seems important to verify to what extent the prognostically positive echocardiographic parameters proposed in this study can be complementary to respiratory function tests [68]. In metastatic NSCLC, functional assessment may also play an important prognostic role even despite the use of the latest therapies [69,70,71]. Even in SCLC, better assessment of functional status and prognostic factors may be helpful in proper planning of treatment strategies [72,73].

Our prospective study shows that echocardiographic parameters may predict survival time in lung cancer patients. This study may become a next important point in the discussion of how echocardiography may be important in predicting all-cause mortality. Our study confirmed that baseline echocardiography could be considered as a prognostic predictor independently of histopathological and molecular diagnosis in inoperable lung cancer. There are other interesting arguments in the literature. For example, Carpeggiani et al. [74] showed that a positive result of stress echocardiography can be a significant predictor of not only cardiovascular mortality but a later cancer specific cause of death. It is important in this study that all patients were cancer-free at the moment stress echocardiography was performed. Therefore, it can be speculated that a positive result indicating contractility disorders in the course of cardiac ischemia also predicts cancer disease development.

Our study revealed many cases of cardiotoxicity in advanced or metastatic lung cancer. Previous studies based on echocardiography in the diagnosis of cardiotoxicity of drugs used in lung cancer indicated a rather low percentage of echocardiographic abnormalities [75,76].

The high incidence of cardiotoxicity in our study resulted from the use of the most recent definition of cardiotoxicity based on the most modern echocardiography with the assessment of GLS. It confirms the need to conduct further studies using modern echocardiographic monitoring of right and left ventricular GLS in patients with locally advanced or metastatic lung cancer, where the prognosis is constantly improving thanks to new anticancer therapies.

## 6. Conclusions

This study supplements current knowledge on prognosis in lung cancer by finding new predictors of OS. Two ECG parameters: atrial fibrillation, heart rate >90/min, and one echocardiographic parameter: high RVSP (>39 mmHg), were identified as negative predictors. The special value of the study is identification of three new echocardiographic determinants for OS: RVSP < 21 mmHg, RV free wall strain <−30%, RV GLS <−25.5%. Additionally, the study showed how low performance status (ECOG = 2) can be characterised by some echocardiographic parameters: RVSP (by each 1 mmHg), ACT (by each 1 ms), as well as by each 1% of RV GLS, RV free wall strain, fractional area—FAC. The proposed diagnostic algorithm may be a key in a new understanding of outcomes in lung cancer.

## Figures and Tables

**Figure 1 cancers-14-02010-f001:**
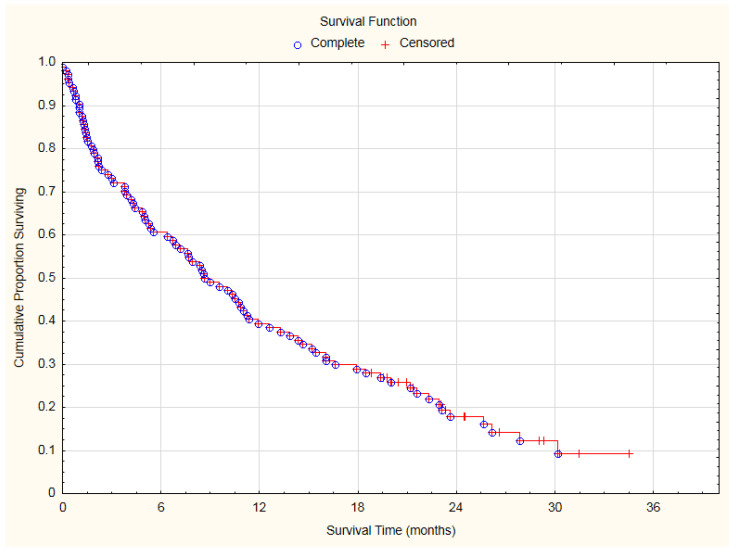
Overall survival curve of analyzed lung cancer patients.

**Table 1 cancers-14-02010-t001:** Characteristics of lung cancer patients included in analyses of survival.

Parameters	Characteristic by Numbers
Sex	
women	29 (27.9%)
men	75 (72.1%)
Age	Mean ± SD: 66.9 ± 8.3
BMI (kg/m^2^)	Mean ± SD: 25.3 ± 5.1
Pathology:	
Non-small-cell lung carcinoma (NSCLC)	
Squamous-cell carcinoma	39 (37.5%)
Adenocarcinoma	33 (31.73%)
Not otherwise specified (NOS)	9 (8.65%)
Large-cell lung carcinoma (LCLC)	1 (0.96%)
Neuroendocrine lung cancer	3 (2.88%)
Small-cell lung carcinoma (SCLC)	19 (18.27%)
Advancement of NSCLC	
metastatic cancer	42 (40.38%)
locally advanced	43 (41.34%)
Advancement of SCLC	
extensive disease	16 (15.38%)
limited disease	3 (2.88%)
Never smokers	16 (15.38%)
Performance status (ECOG, Eastern Cooperative Oncology Group)	
0	25 (24.04%)
1	59 (56.73%)
2	20 (19.23%)
Number of drugs for pain control	
0	35 (33.65%)
1	15 (14.42%)
2	33 (31.73%)
3	18 (17.31%)
4	3 (2.88%)
Comorbidities	
Venous thromboembolic disease	10 (9.62%)
Arterial hypertension	54 (51.92%)
Chronic coronary syndrome	19 (18.27%)
Chronic obstructive pulmonary disease	28 (26.92%)
Hypothyrosis	9 (8.65%)
Chronic renal disease	3 (2.88%)
Diabetes mellitus	14 (13.46%)

**Table 2 cancers-14-02010-t002:** Cardiovascular toxicities of lung cancer therapies diagnosed in accordance with the International Cardio-Oncology Society 2021 consensus statement [26].

Criteria of Diagnosis	Number of Patients	Cancer Therapy
**Echocardiography-Based Events**		
Severe: new LVEF reduction to <40%	4 patients	cisplatin + vinorelbine (1 case), carboplatin + vinorelbine (1 case), carboplatin + etoposide (1 case), pembrolizumab (1 case).
Moderate: new LVEF reduction by ≥10 percentage points to an LVEF of 40–49%: nobody	nobody	-
Moderate: new LVEF reduction by <10 percentage points to an LVEF of 40–49% AND new relative decline in GLS by >15% from baseline	4 patients	cisplatin + vinorelbine (1 case), carboplatin (1 case), pemetrexed (1 case), pembrolizumab (1 case).
Mild: LVEF ≥ 50% AND new relative decline in GLS by >15% from baseline	16 patients	vinorelbine (3 patients), cisplatin + vinorelbine (2 patients), cisplatin + pemetrexed (2 patients), cisplatin + etoposide (2 patients), cisplatin + gemcitabine (2 patients), carboplatin + etoposide (1 case), carboplatin (1 case), pemetrexed (1 case), gemcitabine (1 case), crizotinib (1 case).
**ECG-based events**		
New atrial fibrillation	1 patient	vinorelbine (1 case)
New sinus tachycardia >100/min	3 patients	cisplatin + vinorelbine (2 patients), cisplatin + gemcitabine (1 case),
New QTc > 500 ms	12 patients	cisplatin + vinorelbine (4 patients), cisplatin + etoposide (3 patients), carboplatin + etoposide (1 case), carboplatin (1 case), pemetrexed (1 case), pembrolizumab (1 case). afatinb (1 case)

**Table 3 cancers-14-02010-t003:** Baseline pre-existing clinical oncology and cardiology possible predictors of OS in lung cancer patients. Univariate exploratory analysis.

Possible Predictors		Univariable Analysis		
		HR	95% CI	*p*-Value
Oncology	SCLC vs. NSCLC	1.25	0.74–2.10	0.41
	Diagnosis of Squamous-cell carcinoma	0.77	0.49–1.20	0.24
	Diagnosis of Adenocarcinoma	0.87	0.55–1.38	0.55
	Low Performace Status: ECOG 2 vs. 0/1	2.53	1.51–4.24	0.0004
	Metastatic vs. non-metastatic disease	1.74	1.12–2.68	0.01
	Metastases to central nervous system	1.43	0.81–2.54	0.22
	Metastases to bones	1.75	1.00–3.07	0.049
	Metastases to liver or adrenal glands	2.55	1.60–4.06	0.00008
	Pleural effusion	1.88	1.18–3.00	0.008
	Need for pain relief	1.73	1.08–2.76	0.02
	Each next drug for pain control	1.29	1.08–1.54	0.004
	Morphine for pain control	2.17	1.32–3.56	0.002
	Never-smokers vs. smokers	0.91	0.50–1.64	0.75
	Anemia (HGB < 12 g/dL as lower quartile)	1.55	0.97–2.48	0.07
Cardiology and Internal Medicine	Pericardial effusion	1.49	0.93–2.39	0.1
	Venous thromboembolic disease	0.64	0.29–1.38	0.25
	Arterial hypertension	0.95	0.62–1.44	0.80
	Chronic coronary syndrome	1.45	0.87–2.44	0.16
	ACE/ARB	0.86	0.56–1.31	0.47
	Beta-blokers	1.34	0.86–2.07	0.19
	ASA	1.33	0.85–2.08	0.21
	Statin	1.28	0.81–2.01	0.28
	LMWH/NOAC	1.24	0.54–2.85	0.62
	Atrial Fibrillation in ECG	2.39	1.14–5.03	0.02
	RBBB	1.37	0.71–2.65	0.35
	PQ > 200 ms	0.64	0.26–1.54	0.32
	QTc ≥ 500 ms	0.88	0.47–1.67	0.70
	Heart rate: HR > 90/min	1.67	1.06–2.62	0.03
	Chronic obstructive pulmonary disease	1.38	0.87–2.21	0.17
	Hypothyrosis	0.84	0.41–1.75	0.65
	Chronic renal disease	2.33	0.72–7.49	0.16
	Diabetes mellitus	1.34	0.74–2.43	0.33
	Obesity (BMI ≥ 30 kg/m^2^)	0.8	0.45–1.42	0.44
	Age > 65 years	1.19	0.77–1.84	0.43
	Women vs. men	1.50	0.95–2.36	0.08

**Table 4 cancers-14-02010-t004:** Baseline main parameters of echocardiography as predictors of OS. Values of lower and upper quartiles were adopted as cut-off points. Univariate exploratory analysis.

Echocardiography Parameters		Lower and Upper Quartiles	Univariable Analysis		
			HR	95% CI	*p*-Value
LV diameter and systolic function	LVDD (mm)	<43	1.54	0.95–2.50	0.08
		>52	1.38	0.82–2.32	0.23
	EF (%)	<55	1.27	0.78–2.07	0.34
		>64	1.29	0.78–2.13	0.32
	LV GLS (%)	<−18	1.12	0.66–1.88	0.68
		>−12	1.53	0.94–2.48	0.09
Atrials characteristics	LA (cm^2^)	<14	1.57	0.93–2.65	0.09
		>22	1.44	0.87–2.38	0.16
	RA (cm^2^)	<12	1.36	0.82–2.24	0.23
		>17	1.37	0.85–2.23	0.20
LV diastolic function	FALA E (cm/s)	<60	0.77	0.46–1.28	0.31
		>88	0.95	0.58–1.55	0.84
	E/A	<0.62	0.96	0.58–1.59	0.88
		>0.94	1.34	0.82–2.19	0.24
	E’(cm/s)	<5.55	1.54	0.96–2.48	0.07
		>8.65	1.01	0.63–1.63	0.96
	E/E’	<8	0.73	0.44–1.21	0.22
		>13	1.37	0.84–2.23	0.21
RV function, pulmonary artery pressure and volaemia	TAPSE (mm)	<20	1.39	0.81–2.39	0.24
		>26	1.00	0.59–1.68	1.00
	RVSP (mmHg)	<21	0.36	0.20–0.66	0.0008
		>39	2.01	1.24–3.26	0.0045
	ACT (ms)	<99	1.35	0.84–2.19	0.22
		>135	0.86	0.50–1.49	0.60
	IVC diameter (mm)	<10	1.07	0.65–1.74	0.80
		>17	1.14	0.70–1.85	0.60
	RV GLS (%)	<−25.5	0.59	0.35–1.00	0.05
		>−15	0.94	0.55–1.60	0.81
	RV free wall strain (%)	<−30	0.56	0.32–0.97	0.038
		>−20	0.9	0.54–1.51	0.69
	RV end-diastolic area (cm^2^)	<18.2	1.27	0.78–2.07	0.34
		>25	0.89	0.54–1.47	0.65
	Fractional Area Change -FAC (%)	<39.8	1.18	0.72–1.93	0.52
		>46.8	1.0	0.61–1.65	1.0
	RV S’ (cm/s)	<10	1.61	0.96–2.68	0.07
		>12	1.32	0.79–2.20	0.28

**Table 5 cancers-14-02010-t005:** Diagnosis of low performance status (ECOG 2) in relation to main parameters of echocardiography. Univariate logistic regression.

Echocardiography Parameters		Univariable Analysis		
		OR	95% CI	*p*-Value
LV diameter and systolic function	LVDD (mm)	0.97	0.90–1.05	0.50
	EF (%)	0.97	0.92–1.02	0.17
	LV GLS (%)	1.02	0.94–1.11	0.62
Atrials characteristics	LA (cm^2^)	1.08	0.995–1.17	0.06
	RA (cm^2^)	1.06	0.97–1.17	0.19
LV diastolic function	FALA E (cm/s)	1.00	0.98–1.02	0.94
	E/A	1.47	0.30–7.27	0.63
	E’(cm/s)	0.96	0.80–1.16	0.69
	E/E’	1.02	0.92–1.13	0.66
RV function, pulmonary artery pressure and volaemia	TAPSE (mm)	0.95	0.82–1.09	0.45
	RVSP (mmHg)	1.03	1.01–1.06	0.01
	ACT (ms)	0.97	0.95–0.99	0.01
	IVC diameter (mm)	1.05	0.99–1.11	0.13
	RV GLS (%)	1.14	1.04–1.24	0.003
	RV free wall strain	1.12	1.04–1.22	0.004
	RV end-diastolic area (cm^2^)	1.02	0.92–1.12	0.73
	Fractional Area Change -FAC (%)	0.89	0.82–0.96	0.002
	RV S’ (cm/s)	0.85	0.62–1.18	0.33

## Data Availability

Data may be available upon reasonable request.

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
