# Peer review of "Baseline Electrocardiographic and Echocardiographic Assessment May Help Predict Survival in Lung Cancer Patients—A Prospective Cardio-Oncology Study"

_cancers, 2022, doi:10.3390/cancers14082010_

Round 1

Reviewer 1 Report

Sabina Mędrek et al. investigated “ Baseline Electrocardiographic and Echocardiographic Assessment May Help Predict Survival in Lung Cancer Patients- A Prospective Cardio-Oncology Study, but the following factors should be addressed.

  • Follow-up electrocardiographic and echocardiographic studies
  • Information about chemotherapy treatment
  • The impact of chemotherapy drugs on cardiotoxicity
  • Kaplan-Meier survival curves should be performed

Author Response

Sabina Mędrek et al. investigated “ Baseline Electrocardiographic and Echocardiographic Assessment May Help Predict Survival in Lung Cancer Patients- A Prospective Cardio-Oncology Study, but the following factors should be addressed.

  • Follow-up electrocardiographic and echocardiographic studies

The significant advancement of the lung cancer disease (inoperable stage) resulted in a high mortality. During the first 100 days of the observation, 19 patients (27.88%) died due to cancer-related reasons. The remaining 75 patients underwent electrocardiographic and echocardiographic control. Cardiovascular toxicities of lung cancer therapies were identified according to International Cardio-Oncology Society 2021 consensus statement (Table). Cardiac toxicity was diagnosed in 24 of 75 patients (32%): as severe in 4 patients (5.33%), moderate in the next 4 patients (5.33%), mild in 16 patients (21.33%).  

Table. Cardiovascular toxicities of lung cancer therapies diagnosed in accordance with the International Cardio-Oncology Society 2021 consensus statement.

Number of patients

Cancer therapy

Echocardiography-based events

Severe: new LVEF reduction to <40%

4 patients

cisplatin+vinorelbine (1 case), carboplatin+vinorelbine (1 case), carboplatin+etoposide (1 case), pembrolizumab (1 case).

Moderate: new LVEF reduction by ≥ 10 percentage points to an LVEF of 40-49%: nobody

nobody

-

Moderate: new LVEF reduction by <10 percentage points to an LVEF of 40-49% AND new relative decline in GLS by >15% from baseline

4 patients

cisplatin+vinorelbine (1 case), carboplatin (1 case),

pemetrexed (1 case),

pembrolizumab (1 case).

Mild: LVEF ≥ 50% AND new relative decline in GLS by >15% from baseline

16 patients

vinorelbine (3 patients), cisplatin+vinorelbine (2 patients), cisplatin+pemetrexed (2 patients), cisplatin+etoposide (2 patients), cisplatin+gemcitabine (2 patients), carboplatin+etoposide (1 case),

carboplatin (1 case),

pemetrexed (1 case),

gemcitabine (1 case),

crizotinib (1 case).

ECG-based events

New atrial fibrillation

1 patient

vinorelbine (1 case)

New sinus tachycardia >100/min.

3 patients

cisplatin+vinorelbine (2 patients),

cisplatin+gemcitabine (1 case),

New QTc>500ms

12 patients

cisplatin+vinorelbine (4 patients),

cisplatin+etoposide (3 patients),

carboplatin+etoposide (1 case),

carboplatin (1 case),

pemetrexed (1 case),

pembrolizumab (1 case).

afatinb (1 case)

  • Information about chemotherapy treatment

Qualification for lung cancer treatment was based on the histopathological and molecular diagnosis, taking into account the clinical stage, the performance status and comorbidities including cardiovascular diseases. Finally the patients received at first choice:

  • platinum-based doublet chemotherapy: 72 patients (69.23%)
    • cisplatin-based: 53 patients (50.96%)
      • cisplatin+vinorelbine: 28 patients
      • cisplatin+etoposide: 14 patients
      • cisplatin+pemetrexed: 6 patients
      • cisplatin+gemcitabine: 5 patients
    • carboplatin-based: 19 patients (18.27%)
      • carboplatin+etoposide: 10 patients
      • carboplatin+vinorelbine: 8 patients
      • carboplatin+paklitaksel: 1 patient
    • cytotoxics monotherapy: 17 patients (16.35%)
      • pemetrexed: 6 patients
      • vinorelbine: 5 patients
      • carboplatin: 3 patients
      • gemcitabine: 2 patients
      • taxanes: 1 patient
    • monotherapy with pembrolizumab: 8 patients (7.69%)
    • targeted therapy: 3 patients (2.88%): two with EGFR inhibitor and one with ALK inhibitor
    • alone radiotherapy: 2 patents (1.92%)
    • best supportive care: 2 patients (1.92%).

Additionally 6 patients (5.77%) received radiotherapy due to locally advanced NSCLC as part of chemoradiotherapy (concomitant in 2 patients, sequential in 4 patients). The next 6 patients (5.77%) were treated by radiation therapy due to metastases to bone or central nervous system.

In the long-term follow-up when lung cancer disease progression was recognized as second line of cancer therapy were recommended:

  • chemotherapy: 12 patients (11.54%) including: platinum-based scheme in 6 patients, gemcitabine in 3 patients, taxanes in 2 patients and CAV chemotherapy (cyclophosphamide, doxorubicin and vincristine) in one patient 
  • immune checkpoint inhibitors: 4 patients (3.85%) including: atezolizumab in 2 patients, nivolumab in one patient, pembrolizumab in one patient
  • radiation therapy in 33 patients (31.73%): for lung in 16 patients, for central nervous system in 9 patients, for mediastinum in 5 patients, for other site of metastasis in 3 patients.

  • The impact of chemotherapy drugs on cardiotoxicity

The impact of anticancer therapy on different types of cardiotoxicity was presented in the table. Details are shown as individual cases because of the variety of anti-cancer therapy used.

  • Kaplan-Meier survival curves should be performed

During the study observation 87 patients (83.65%) died. There were no clearly confirmed cardiovascular reasons of death. At the end of follow-up 17 patients (16.35%) were alive. Median of overall survival (OS) was 260 days (8.67 months) with interquartile range: 71 - 623 days ( 2.37 - 20.77 months). The 1-year and 2-year survival rates were 39.42% and 17.84%, respectively (Figure).  

Reviewer 2 Report

Medrek et al conducted a prospective analysis of lung cancer patients to determine prognostic value of ECG and echocardiography in patient survival. 

  1. Introduction: The authors should revise the introduction. The description of risk factors in cardiovascular diseases and cancer is not well structured and confusing. The description of underlying cardiovascular complications or diseases in lung cancer patients needs to be expanded. The authors should also consider to add information and definitions of lung cancer types which are relevant for the present study (e.g., NSCLC vs SCLC). The definition on heart failure needs more details.
  2. Method section: The authors need to provide more details on study design, subject/patient selection process, age ranges, end points and inclusion, as well exclusion criteria. Further, settings and locations where the data were collected, sample size calculation, 
  3. Statistical analysis: Statistical methods used to compare groups and software used for data analysis.
  4. Arial fibrillation was identified as a negative predictor in lung cancer patients. The patient profile shows that ~3/4 of individuals were male, but echocardiography data is reported without any sex differences. AF has a higher prevalence in male individuals independent of any cancer diagnosis compared to females. Did the authors test for sex differences? How did the authors treat sex and age in their analysis of cardiac function?
  5. Atrial fibrillation is frequently observed in response to cancer therapies. The authors need to include details on therapy type and duration for included patients. The authors should also discuss if there is a correlation between tumor type, therapy and the onset of atrial fibrillation?

Author Response

Medrek et al conducted a prospective analysis of lung cancer patients to determine prognostic value of ECG and echocardiography in patient survival. 

  1. Introduction: The authors should revise the introduction. The description of risk factors in cardiovascular diseases and cancer is not well structured and confusing. The description of underlying cardiovascular complications or diseases in lung cancer patients needs to be expanded. The authors should also consider to add information and definitions of lung cancer types which are relevant for the present study (e.g., NSCLC vs SCLC). The definition on heart failure needs more details.

We have improved the quality of the Introduction by:

- clear presentation of relationship between cardiovascular disease and cancer

- expanded description of concomitant cardiovascular diseases in lung cancer patents

- supplementation of definitions of lung cancer and heart failure

  1. Method section: The authors need to provide more details on study design, subject/patient selection process, age ranges, end points and inclusion, as well exclusion criteria. Further, settings and locations where the data were collected, sample size calculation, 

We have extended the method section by:

The study was designed as a pilot study to test the potential relationship between overall mortality and echocardiographic or electrocardographic changes related to advanced lung cancer disease. The idea of the study was to check whether echocardiographic and electrocardiographic parameters enhanced by lung cancer development could complement the prognostic risk scales in inoperable advanced or metastatic lung cancer disease.

The study was conducted in one cancer center that specializes in treating only advanced / metastatic lung cancer disease. Due to the fact that the center does not have thoracic surgery, patients with surgical treatment are referred to another center.

Each patient enrolled in the study had to be disqualified from lung cancer surgery by a multidisciplinary team.

The objective endpoint of the study was all-cause mortality, therefore it was possible to include patients with various histopathological and molecular diagnoses.

It was planned to include at least 100 patients so that the demonstrated relationships become the basis for creating a study dedicated to particular types of advancement (III vs IV) and typical histopathological (at least SCLC vs NSCLC) and a specific treatment (radical chemoradiotherapy in locally advanced disease, immunotherapy and chemotherapy in metastatic disease). etc.)

  1. Statistical analysis: Statistical methods used to compare groups and software used for data analysis.

All statistical analyses were performed using STATISTICA software.

  1. Arial fibrillation was identified as a negative predictor in lung cancer patients. The patient profile shows that ~3/4 of individuals were male, but echocardiography data is reported without any sex differences. AF has a higher prevalence in male individuals independent of any cancer diagnosis compared to females. Did the authors test for sex differences? How did the authors treat sex and age in their analysis of cardiac function?

Due to the fact that atrial fibrillation is more common in men and in elderly patients, we carefully checked who was diagnosed with atrial fibrillation at baseline in our population (please see the text).

In the analysis of prognostic factors, we took into account age and gender (please see Table 3).

We added to the limitations of the study that due to the small size of the pilot study, we did not apply the echocardiographic parameters to age and gender.

  1. Atrial fibrillation is frequently observed in response to cancer therapies. The authors need to include details on therapy type and duration for included patients. The authors should also discuss if there is a correlation between tumor type, therapy and the onset of atrial fibrillation?

Many thanks for this comment, we have added the information:

Atrial fibrillation can be observed in response to anticancer therapies, but in our population only one patient - 75 years old woman with history of arterial hypertension and chronic pulmonary disease, with low performance status ECOG 2, experienced atrial fibrillation during treatment with vinorelbine due to lung adenocarcinoma (Table).

Reviewer 3 Report

In this prospective study the authors examined the echocardiographic
parameters of lung cancer patients with their survival time.
Previous reports suggested the use of echocardiography in cancer 
patients as predictor of mortality such as Carpeggiani et al.
(10.1161/JAHA.117.007104), as well as in lung cancer in particular
Omersa et al. (10.1515/raon-2016-0037) and Mulliez et al. 
(10.1016/j.radonc.2020.03.022). So, the submitted study, adds to
the existing publication record.

Concerns
1. It seems that all patients examined suffer from advanced
lung cancer. Is this correct? In the materials and methods it is
stated that they had either metastatic cancer or locally advanced
cancer not eligible for surgery. A common feature in many such
cases is that the cancer is covering of the heart (pericardium), 
which could compromise heart performance. How many of the cases
had lung cancer spread to the heart?
2. Half of the cases studied suffer from arterial hypertension.
Was this condition precede cancer? In how many cases arterial 
hypertension precede lung cancer?
3. The end points of the study should be stated clearly in methods. 
Also, the patients mortality should be defined as cancer-related,
cardiac-related and all-cause mortality. 
4. The survival time of patients should be expressed through 
Kaplan-Meyer survival curves.
5. The authors should discussed the use of echocardiography
in cancer as predictor of patient mortality and compare their
findings with the results of previous reports.

Author Response

In this prospective study the authors examined the echocardiographic
parameters of lung cancer patients with their survival time.
Previous reports suggested the use of echocardiography in cancer 
patients as predictor of mortality such as Carpeggiani et al.
(10.1161/JAHA.117.007104), as well as in lung cancer in particular
Omersa et al. (10.1515/raon-2016-0037) and Mulliez et al. 
(10.1016/j.radonc.2020.03.022). So, the submitted study, adds to
the existing publication record.

Concerns
1. It seems that all patients examined suffer from advanced
lung cancer. Is this correct? In the materials and methods it is
stated that they had either metastatic cancer or locally advanced
cancer not eligible for surgery. A common feature in many such
cases is that the cancer is covering of the heart (pericardium), 
which could compromise heart performance. How many of the cases
had lung cancer spread to the heart?

According to the inclusion criteria, all patients had advanced or metastatic lung cancer disease. Each patient had an initial CT scan to objectively assess the advancement of the neoplastic disease. In none of the patients the tomography showed that the myocardium or pericardium was directly affected by the neoplasm, because then the prognosis would be much worse due to the direct involvement of the heart structures.

  1. Half of the cases studied suffer from arterial hypertension.
    Was this condition precede cancer? In how many cases arterial 
    hypertension precede lung cancer?

All cases of arterial hypertension were recognized prior to the diagnosis of lung cancer. After the initiation of anticancer treatment, there was no new diagnosis of arterial hypertension, but there were 7 patients who were discontinued ACEI or ARB due to the observed hypotension.

  1. The end points of the study should be stated clearly in methods. 
    Also, the patients mortality should be defined as cancer-related,
    cardiac-related and all-cause mortality. 

The end-point of the study was all-cause mortality.

  1. The survival time of patients should be expressed through 
    Kaplan-Meyer survival curves.

We have added the Kaplan-Meyer survival curve (Figure) and data with survival time (in the text).

  1. The authors should discussed the use of echocardiography
    in cancer as predictor of patient mortality and compare their
    findings with the results of previous reports.

Our prospective study shows that echocardiographic parameters may predict survival time in lung cancer patients. This study may become a next important point in the discussion of how echocardiography may be important in predicting all-cause mortality. Our study confirmed that baseline echocardiography could be considered as prognostic predictor independently of histopathological and molecular diagnosis in inoperable lung cancer. There are other interesting arguments in the literature. For example Carpeggiani et al. showed that positive result of stress echocardiography can be a significant predictor of not only cardiovascular mortality but later cancer specific cause of death. It is important in this study that all patients were cancer-free at the moment of stress echocardiography performing. Therefore, it can be speculated that a positive result indicating contractility disorders in the course of cardiac ischemia also predicts cancer disease development.

Our study revealed many cases of cardiotoxicity in advanced or metastatic lung cancer. Previous studies based on echocardiography in the diagnosis of cardiotoxicity of drugs used in lung cancer indicated a rather low percentage of echocardiographic abnormalities [Wachters et al., Omersa et al.]

The high incidence of cardiotoxicity in our study resulted from the use of the most recent definition of cardiotoxicity based on the most modern echocardiography with the assessment of GLS. It confirms the need to conduct further studies using modern echocardiographic monitoring of right and left ventricular GLS in patients with locally advanced or metastatic lung cancer, where the prognosis is constantly improving thanks to new anticancer therapies.

New references:

Carpeggiani C, Landi P, Michelassi C, Andreassi MG, Sicari R, Picano E. Stress Echocardiography Positivity Predicts Cancer Death. J Am Heart Assoc. 2017 Dec 12;6(12):e007104. 

Wachters FM, Van Der Graaf WT, Groen HJ. Cardiotoxicity in advanced non-small cell lung cancer patients treated with platinum and non-platinum based combinations as first-line treatment. Anticancer Res. 2004 May-Jun;24(3b):2079-83.

Omersa D, Cufer T, Marcun R, Lainscak M. Echocardiography and cardiac biomarkers in patients with non-small cell lung cancer treated with platinum-based chemotherapy. Radiol Oncol. 2016 Jun 24;51(1):15-22.

Round 2

Reviewer 1 Report

Thank you for considering my suggestions. 

Reviewer 2 Report

The authors have extensively revised the manuscript and addressed all my previous concerns.

Reviewer 3 Report

In this prospective study the authors examined the echocardiographic
parameters of lung cancer patients with their survival time.
It is an important contribution with clinical interest.